# Short email with attachment versus long email without attachment when contacting authors to request unpublished data for a systematic review: a nested randomised trial

Peter J Godolphin,[1,2] Philip M Bath,[2] Alan A Montgomery[1]

¹Nottingham Clinical Trials Unit, The University of Nottingham, Nottingham, UK
²Stroke Trials Unit, Division of Clinical Neuroscience, University of Nottingham, Nottingham, UK

**Correspondence to**
Mr Peter J Godolphin;
peter.godolphin@nottingham.ac.uk

## ABSTRACT

**Objective** Systematic reviews often rely on the acquisition of unpublished analyses or data. We carried out a nested randomised trial comparing two different approaches for contacting authors to request additional data for a systematic review.

**Participants** Participants were authors of published reports of prevention or treatment trials in stroke in which there was central adjudication of events. A primary and secondary research active author were selected as contacts for each trial.

**Interventions** Authors were randomised to be sent either a short email with a protocol of the systematic review attached ('Short') or a longer email that contained detailed information and without the protocol attached ('Long'). A maximum of two emails were sent to each author to obtain a response. The unit of analysis was trial, accounting for clustering by author.

**Primary and secondary outcome measures** The primary outcome was whether a response was received from authors. Secondary outcomes included time to response, number of reminders needed before a response was received and whether authors agreed to collaborate.

**Results** 88 trials with 76 primary authors were identified in the systematic review, and of these, 36 authors were randomised to Short (trials=45) and 40 to Long (trials=43). Responses were received for 69 trials. There was no evidence of a difference in response rate between trial arms (Short vs Long, OR 1.10, 95% CI 0.36 to 3.33). There was no evidence of a difference in time to response between trial arms (Short vs Long, HR 0.91, 95% CI 0.55 to 1.51). In total, 27% of authors responded within a day and 22% of authors never responded.

**Conclusions** There was no evidence to suggest that email format had an impact on the number of responses received when acquiring data for a systematic review involving stroke trials or the time taken to receive these responses.

## Strengths and limitations of this study

► This is the first randomised trial comparing different email formats when contacting authors to request unpublished data for a systematic review.
► This study follows a clear and rigorous protocol, guided by experienced methodologists, and implemented in a clinical trials unit.
► Blinding was not possible as it was clear which format of email had been sent. However, contacted authors were not aware of this trial during the study, and therefore, we expected that their response rate and time to response would remain unbiased.
► The sample size is constrained by the number of eligible studies identified for the systematic review. It is possible that a larger study would detect small but important effects of email format and presence or absence of an attachment on response rate.

## INTRODUCTION

Systematic reviews are often considered as the highest level of evidence available.[1] However, meta-analyses, the statistical component of a systematic review, often rely on the acquisition of unpublished summary results or further data. It can be a challenge to obtain this data, especially if the meta-analysis requires individual patient data,[2,3] resulting in insufficient studies to pool together.[4]

Data sharing is not yet common practice in the field of biomedical research, and many researchers struggle to acquire the underlying data sets used in journal articles.[5] There are many reasons why there may be difficulties in acquiring data; authors may not wish to share their data,[6] they may be too busy to respond or deal with requests,[4] they may have moved institution and be uncontactable, the data could have been lost or destroyed,[7] or authors may not have access to the data.[7] Author inclination to share data could be related to the strength of evidence and the perceived quality of statistical results from their study.[8] Research studies that report borderline evidence of an intervention effect may be less likely to take part in data

sharing, which would result in publication bias for a systematic review.[9] Various methods are currently employed to encourage authors to share data for a systematic review, and a randomised trial investigating if financial incentives may be effective is currently under way.[10] Additionally, academic incentives for making data available could be introduced, such as invitation into group collaboration, or having data sharing recognised to an equivalent level as scientific presentations or journal publications.[11]

There appears to be no consensus about the best way to approach authors to contact them for data. The corresponding author should be the most appropriate person to contact when attempting to acquire unpublished clinical trial data, but contacting these authors may be challenging as they could be inaccessible or have little time to deal with requests.[12] Therefore, if email contact is used, this must be clear, concise and easy to respond to. Some authors may prefer a shorter email, with additional information attached, while others may prefer a longer email with sufficient information they need to make a decision without reading additional attached information such as a study protocol.

The initial interaction with any author may have a large impact on whether they collaborate, or even on whether they reply at all. The first step when attempting to acquire data is to open up a channel of communication with the data custodian, assumed in this paper to be the author of the journal article. Thus, an important outcome to assess is whether an initial response is received, before considering if interaction with the author affects willingness to share data.

We carried out a nested randomised trial, which investigated two different approaches to contacting authors for data in a systematic review, with the primary aim of seeing which method elicits the most complete response.

## METHODS
### Participants
The systematic review chosen for this study investigates whether central adjudication of the primary outcome in stroke trials has any impact on the main trial primary analysis.[13] Since the information required for the review is not commonly published in the main or secondary trial publications, author contact is essential which makes it a useful setting to test the effect of different ways of contacting authors on response rate.

In brief, the studies included were those adjudged to be randomised trials of prevention or treatment of stroke, which had centrally adjudicated their primary outcome. MEDLINE, Embase, the Cochrane Central Register of Controlled Trials, Web of Science, PsycINFO and Google Scholar were searched for relevant articles. We restricted searches from Google Scholar to the first 300 articles,[14] and only articles written in English were considered.

Two author contacts were chosen from each identified trial, and these were the participants selected for this study. The primary contact was ideally one of the corresponding authors. This author was checked on PubMed to identify whether they were still an active researcher (paper published within the last two years). If the corresponding author appeared to have ceased research activity, then a secondary author who was still an active researcher (preferably first/second/last) was selected as the primary point of contact. Additionally, a second research active author was chosen as a second contact for the trial. Ideally, this second contact also had a major role in the trial, determined by their position on the main trial publication and in the author contribution section of the manuscript, if completed.

Authors were invited to collaborate in our research programme investigating adjudication in stroke trials. In order to collaborate, they were asked to provide data in one of two formats: individual participant data or summary results. Authors that provided data were given the opportunity to critically review the findings and draft manuscript. Those that accepted this offer were invited to join the systematic review writing committee.

### Patient and public involvement
Patients were not involved in this study.

### Trial design
This study was a parallel group, randomised trial of short email and attachment versus long email to elicit response from potential collaborators. The trial was nested within a systematic review and meta-analysis. Blinding was not possible, as it was clear to contacted authors and the trial team which format of email had been sent. However, contacted authors were unaware that alternative formats of contact email were being compared in a nested randomised study, and therefore, we expected that their willingness to respond would be unbiased.

### Randomisation
Randomisation occurred at author level. This was to ensure that if any author had published multiple trials that were to be included, then they only received one email, rather than multiple emails, which could be in different formats. Randomisation was stratified by year of publication (median split: <2011/≥2011), size of trial (median split: <1738/≥1738) and if the author had multiple trials which are included (yes/no). Therefore, there were eight strata. If authors had multiple trials, then the within-author median size and median year were used when stratifying. Allocation was concealed and a researcher not involved in the trial randomised participants using a computer-generated random number list.

Participants were randomised in a 1:1 ratio to either of the two treatment groups: (1) Short, where they received a short email message that fits on one screen without the need to scroll, with the protocol attached (Intervention group); (2) Long, where they received a longer email message that contained a fuller description of the systematic review and request for information, with the protocol available on request (Comparator group). The protocol

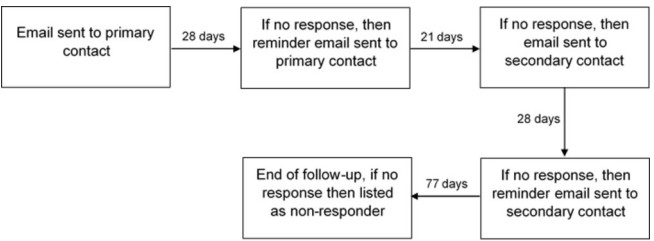

**Figure 1** Plan to elicit response.

detailed the purpose of the systematic review, the data that was requested and the planned programme of research (see online supplementary appendices A and B).

### Plan for eliciting response

Each author received a maximum of two emails. The primary contact was first sent an email in the format of the randomly allocated intervention. If no response was received, then a second email was sent 28 days later, which included some additional brief reminder text, which was the same for both groups, added at the top of the original email (achieved using Reply to all in Outlook).

If there was still no response from the primary contact author after a further 21 days, the second contact author was sent an email in an identical format, waiting 28 days before a reminder was sent. The second contact was not emailed if the first contact responded within the 7-week time frame. Therefore, there was a maximum of four emails in total that authors received if there was no response. Participants were followed up for a further 11 weeks after the fourth and final email. After this stage, any authors that had not responded were recorded as non-respondents, regardless of contact thereafter (figure 1).

### Outcome measures

The primary outcome for this nested randomised trial was whether a response was received from either of the two contacted authors in the trial time frame. This was a binary outcome (response received, yes/no). Out-of-office automatic replies were not counted as a response, and no additional emails were sent on receipt of an out-of-office reply.

Secondary outcomes were time to response, measured in days and the number of emails needed to respond. On receipt of an initial response, this was categorised as either being negative, neutral or positive. Positive responses were allocated if there was an interest in the collaboration and research questions, above simply asking for further information. Negative responses were assigned if participants gave the indication that they did not want to be involved/did not have time to take part. All other responses were coded as neutral.

The eventual outcome on the end of the trial time frame was allocated as either (1) agree to collaborate, (2) do not agree to collaborate, (3) no decision reached or (4) no response. If there was an agreement in principle regardless of whether data had been received, then this was assigned as agree. If there was a clear indication that the author could not/did not want to help, then this was allocated do not agree. No decision reached was given if it was still unclear whether the author would be willing to take part in the collaboration.

### Statistical analysis

The sample size for this nested randomised trial was fixed by the number of studies identified in the systematic review, and although allocation was by author, the unit of analysis is the individual trial. A sample size of 88 trials could detect a difference in response rate between groups of ≤30 percentage points (equivalent OR 3.4) with 80% power and 5% two-sided alpha, ignoring any reduction in effective sample size due to the clustering effect of authors with multiple studies. The characteristics of each trial were summarised using appropriate descriptive statistics for each intervention arm. The primary approach to between-group comparative analyses was by intention-to-treat. Short was treated as the intervention group and Long treated as the comparator group for all between-group analyses. The evaluation of response rate was performed using adjusted logistic regression models. The adjusted model included stratification factors; year of trial publication, trial sample size and multiple trials per author. The primary efficacy parameter comparing interventions was the OR along with the corresponding 95% CI. Robust SEs were used to account for correlation between multiple trials for the same author. A sensitivity analysis was carried out by further adjusting the primary analysis model for any characteristics of trials with marked imbalance between intervention groups.

Time to response was investigated using survival analysis, with response being the outcome of interest. Adjusted HRs, adjusted for stratification factors, were reported alongside 95% CIs. Additional secondary outcomes were summarised using mean, SD, median, lower and upper quartiles, or frequency counts and percentages where appropriate. All analyses were performed in Stata V.15.0 or later.

### RESULTS

The study commenced on 11 July 2017. Eighty-eight trials were identified from a systematic search of the literature and were to be included in the review. This corresponded to 76 unique authors who were selected as the primary contact for these studies. In total, 36 authors were randomised to Short (45 trials) and 40 to Long (43 trials) email formats. The majority of the studies had their main publication published between 2006 and 2015, were either primary or secondary prevention stroke trials, were carried out in >50 trial centres and had a median number of patients randomised >1000 (table 1). Trials randomised to each group had similar study design, intervention type and comparator, but studies allocated to Long tended to have more participants included and a larger number of trial centres. The number of intervention groups was similar for studies allocated to either Short or Long.

**Table 1** Characteristics of included trials

| | Short message and protocol (n=45) | Long message (n=43) |
|---|---|---|
| Number of authors randomised | 36 | 40 |
| Trials per author* | | |
| 1 | 30 (83%) | 37 (93%) |
| 2 | 4 (11%) | 3 (8%) |
| 3 | 1 (3%) | 0 (0%) |
| 4 | 1 (3%) | 0 (0%) |
| Year of main trial publication | | |
| 1990–2000 | 5 (11%) | 2 (5%) |
| 2001–2005 | 7 (16%) | 3 (7%) |
| 2006–2010 | 11 (24%) | 11 (26%) |
| 2011–2015 | 19 (42%) | 22 (51%) |
| 2016–2017 | 3 (7%) | 5 (12%) |
| Patients randomised | | |
| Mean [SD] | 3910.8 [5593.4] | 3755.7 [5154.8] |
| Median [25th, 75th centile] | 1224 [439, 5170] | 1809 [500, 4576] |
| Min, max | 74, 20332 | 48, 21105 |
| Type of trial | | |
| Primary prevention | 20 (44%) | 16 (37%) |
| Secondary prevention | 19 (42%) | 21 (49%) |
| Acute stroke | 6 (13%) | 6 (14%) |
| Setting | | |
| 1 continent | 30 (67%) | 17 (40%) |
| >1 continent | 14 (31%) | 23 (53%) |
| Not found | 1 (2%) | 3 (7%) |
| Number of centres | | |
| Mean [SD] | 212.8 [286.7] | 165.5 [293.4] |
| Median [25th, 75th centile] | 67 [27, 260] | 85 [32, 141] |
| Min, max | 1, 1034 | 4, 1393 |
| Study design | | |
| Parallel† | 44 (98%) | 41 (95%) |
| Factorial | 1 (2%) | 2 (5%) |
| Intervention | | |
| Drug | 30 (67%) | 31 (72%) |
| Surgery/procedure | 12 (27%) | 11 (26%) |
| Device | 2 (4%) | 1 (2%) |
| Diet | 1 (2%) | 0 (0%) |
| Comparator | | |
| Placebo | 6 (13%) | 10 (23%) |
| Standard care | 28 (62%) | 24 (56%) |
| Active treatment | 8 (18%) | 6 (14%) |
| Surgery/procedure | 3 (7%) | 3 (7%) |
| Number of intervention groups | | |
| 2 | 39 (87%) | 38 (88%) |
| 3 | 4 (9%) | 3 (7%) |

Continued

**Table 1** Continued

| | Short message and protocol (n=45) | Long message (n=43) |
|---|---|---|
| 4 | 1 (2%) | 1 (2%) |
| >4 | 1 (2%) | 1 (2%) |
| Primary contact | | |
| First author | 32 (71%) | 30 (70%) |
| Second author | 2 (4%) | 3 (7%) |
| Last author | 4 (9%) | 5 (12%) |
| Other | 7 (16%) | 5 (12%) |
| *Corresponding author* | 32 (71%) | 37 (86%) |
| Secondary contact | | |
| First author | 5 (11%) | 7 (16%) |
| Second author | 8 (18%) | 9 (21%) |
| Last author | 18 (40%) | 17 (40%) |
| Other | 14 (31%) | 10 (23%) |
| *Corresponding author* | 4 (9%) | 3 (7%) |

Data are n (%) unless otherwise specified.
*Unit of allocation was carried out at author level; therefore, number of trials per author do not sum to total N.
†Two parallel trials were cluster randomised (Short=1; Long=1).

The majority of authors chosen as the primary contact were the first author of the main publication or listed first in the writing committee if the article was published under a collaborative group. Conversely, the last author was the most common choice for the second contact, although there was greater variation in choice for this contact, with a position other than first, second or last also prevalent. For studies allocated to Long, the primary contact was more likely to be the corresponding author compared with trials allocated to Short, although author position was similar for both groups (table 1).

In all, responses were received for 69 trials, with authors of 19 studies (22%) never responding. There was no evidence of a difference in response rate between trial arms (table 2 and online supplementary appendix C). Average time to response was marginally quicker among those allocated to Short, with a greater proportion of authors in this trial arm responding to the first email, but there was insufficient evidence to determine a difference between trial arms on time to response (table 3 and figure 2). In total, 27% of authors responded within a day and 38% replied within a week.

**Table 2** Primary outcome: response from contacted authors

| | Response | Adjusted OR (95% CI) | P value |
|---|---|---|---|
| Short message and protocol | 36/45 (80%) | 1.10 (0.36 to 3.33) | 0.87 |
| Long message | 33/43 (77%) | | |

Adjusted for year of publication, size of trial and if the author had multiple trials included. Total N included=88. Robust SEs were used in model fitting.

**Table 3** Time to response (days)

| | Time to response | | | |
|---|---|---|---|---|
| | **Mean [SD]** | **Median [25th, 75th centile]** | **Adjusted HR (95% CI)** | **P value** |
| Short message and protocol | 51.6 [64.7] | 28 [1, 77] | 0.91 (0.55 to 1.51) | 0.72 |
| Long message | 60 [65.3] | 42 [1, 77] | | |

Adjusted for year of publication, size of trial and if the author had multiple trials included. Total N included=88.

A short email elicited a more favourable first response, with few negative replies. However, this did not lead to a larger number of authors agreeing to collaborate, with many authors not following up on their positive intent. Conversely, those allocated the longer email were more likely to respond in a negative fashion, but this meant that the decision not to collaborate was established sooner, and there was a smaller proportion with no decision reached (table 4).

## DISCUSSION

This nested randomised trial found no evidence that email format or presence or absence of an attachment had an impact on author response when attempting to acquire data for a systematic review. Authors responded in a similar time frame and needed a similar number of reminders before a response was received. There was some indication that authors that received a shorter email were more likely to respond positively at first, but there was no evidence that email format changed the likelihood of collaboration.

Approximately three-quarters of authors in the study responded regardless of email format. A study which attempted to contact authors to obtain data for a diagnostic accuracy review for hepatic fibrosis found a similar response rate, with 68% of authors responding to requests for data.[15] For both email formats we included text which invited authors who provided data into a collaboration to assist (at their discretion) with interpretation and write-up of results, and co-authorship as part of a collaborative group. Our view was that offering greater opportunity for

collaboration to authors rather than simply providing data or further results might produce a greater and more complete response. Offering collaboration may have influenced overall response rate, and thus the results of this study may not be generalisable to others where collaboration is not offered. For example, a study which aimed to establish whether corresponding authors accepted responsibility of correspondence, where collaboration was not offered, found a far lower response rate (190/446, 43%).[12] A qualitative interview study which investigated strategies to access unpublished clinical trial data mentioned a lack of collaboration as a main barrier to data sharing.[16] Thus, offering the opportunity for authors to collaborate and being upfront about giving collaborators the chance to input in both interpreting results and the write-up of any papers could give a more positive and thorough response.

**Table 4** Secondary outcomes

| | Short message and protocol (n=36) | Long message (n=40) |
|---|---|---|
| **Number of emails before response received** | | |
| 1 | 16 (44%) | 15 (38%) |
| 2 | 6 (17%) | 10 (25%) |
| 3 | 3 (8%) | 3 (8%) |
| 4 | 2 (6%) | 2 (5%) |
| No response | 9 (25%) | 10 (25%) |
| Mean [SD] | 1.7 [1.0] | 1.7 [0.9] |
| Median [25th, 75th centile] | 1 [1, 2] | 1.5 [1, 2] |
| Min, max | 1, 4 | 1, 4 |
| **First response outcome** | | |
| Negative | 2 (6%) | 8 (20%) |
| Neutral | 9 (25%) | 14 (35%) |
| Positive | 16 (44%) | 8 (20%) |
| **Eventual outcome** | | |
| Agree to collaborate | 11 (31%) | 10 (25%) |
| Do not agree to collaborate | 3 (8%) | 11 (28%) |
| No decision reached | 13 (36%) | 9 (23%) |
| No response | 9 (25%) | 10 (25%) |

Data are n (%) unless otherwise specified. Data are presented on author level, rather than trial level.

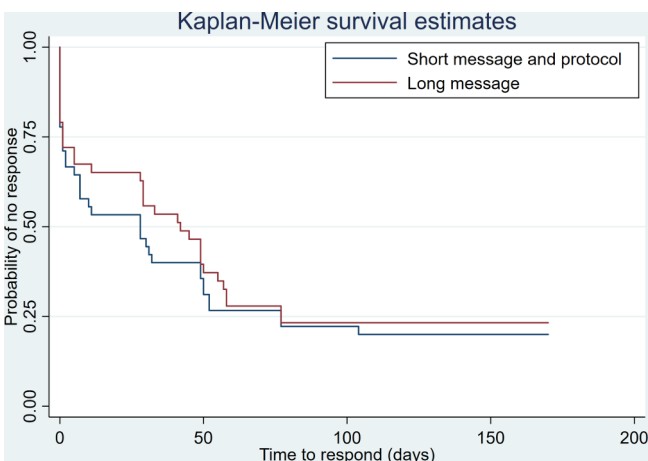

**Figure 2** Kaplan-Meier curve for time to response for both intervention groups.

While short emails elicited a marginally quicker response on average, there was insufficient evidence to conclude benefit of this email style on time to response. Selph et al[15] found that responses were received from 23% of authors after one request, with 55% responding after just two requests (one reminder). This is similar to our study, where we found that 41% of authors had responded after one email and two emails (one reminder) elicited 62% response rate. A third email provided 70% response rate, with little further improvement following a fourth email. The number of emails sent and the number sent to each contact (if there are multiple like in our study) are additional areas where similar research could help to introduce a more rigorous evidence base.

There was some indication that authors allocated to a short email format were more likely to respond positively, with only two negative first responses received from these authors, and double the number of positive responses. However, the number of authors eventually agreeing to collaborate was similar for both groups. This suggests that while the long email format resulted in more negative initial responses, authors who did respond were aware of the implications of our study and could make an informed decision. Thus, the majority of positive first responses to those allocated to the long email format turned into collaboration. It could be hypothesised that while those allocated to the short email format had a detailed protocol attached, authors may not have opened attachments and therefore were agreeing to be involved in the study without fully realised the commitment. This led to a far larger amount of time and resource dealing with these authors, who eventually declined to collaborate, or no decision was reached in the trial time frame.

A limitation of this study was the constrained sample size, enforced by the systematic within which this study was nested. Thus, power to detect a small difference in response rate was low. However, this study does provide a framework to carry out a larger trial of this kind. A further limitation of our trial was that blinding was not possible because the investigator was responsible for sending all emails and collating the data. It was impossible to mask which intervention participants were assigned to due to the email thread being used to reply to messages and send reminders. Therefore, there is a possibility that some outcomes are at a risk of bias. Furthermore, the eventual outcome was not expected to be greatly influenced by the intervention and was more reliant on subsequent communication such as emails or phone contact. However, the protocol was finalised in advance of randomisation, and the trial team followed this protocol, therefore the risk of bias from the trial team was low. In addition, study authors, although aware of email format they received, were not aware of the alternative email format nor were they aware of the nested randomised study, therefore the risk of bias from participants was low.

It could be argued that this study did not investigate what could be perceived as the key question, whether data were shared. This was due to our belief that whether authors provided data was dependent on other factors, and not the original email length and style. Our study investigated an intervention which allows researchers to open up a channel of communication with the data custodian, and there is the potential to use a different intervention to evaluate how best to acquire data (eg, financial incentives[10]).

This nested randomised trial found no evidence that email length had an impact on author response, or agreement to collaborate, when attempting to acquire data in our systematic review. Further studies could test similar hypotheses to evaluate this further, with a greater sample size and in a different clinical area. Given this issue involves countless researchers who strive to generate high-quality research through systematic reviews, this problem seems one that is not only important to tackle, but could be potentially simple to answer. While the first question has to be whether an author responds, the more important question to answer is whether or not an author agrees to collaborate, and arguably even more important, whether or not data are shared. These questions go far beyond the scope of this small trial, but future studies building on this could investigate these issues and attempt to find an intervention that not only boosts response, but also boosts collaboration.

**Contributors** All authors conceived the study. PJG wrote the analysis plan. AAM and PMB reviewed the analysis plan. PJG collected the data, carried out interventions and undertook data analysis. PJG, AAM and PMB interpreted the data. PJG wrote the first draft of the manuscript. AAM and PMB commented on successive versions of the manuscript for important intellectual content. All authors approved the final version of the manuscript.

**Funding** PJG was funded for this summary of independent research by the National Institute for Health Research (NIHR)'s Doctoral Research Fellowship Programme (DRF-2016-09-057). The views expressed are those of the authors and not necessarily those of the NHS, the NIHR or the Department of Health. PMB is Stroke Association Professor of Stroke Medicine, and is a NIHR Senior Investigator.

**Competing interests** None declared.

**Patient consent for publication** Not required.

**Ethics approval** As this was a randomised trial of a methodology rather than a treatment involving patients, we considered that prior consent and trial registration were not appropriate. Furthermore, the first contact with participants had to be randomised otherwise this would negate the purpose of this study.

**Provenance and peer review** Not commissioned; externally peer reviewed.

**Data sharing statement** The data sets generated during this study will be available on request from Peter Godolphin (peter.godolphin@nottingham.ac.uk), a minimum of 6 months after publication. Access to the data will be subject to a review of a data sharing and use request by a committee, and will only be granted upon receipt of a data sharing and use agreement. Any data shared will be anonymised which may affect the reproducibility of published analysis.

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
