## [Reviewer comments · BMJ Open]

This paper was submitted to a another journal from BMJ but declined for publication following peer review. The authors addressed the reviewers' comments and submitted the revised paper to BMJ Open. The paper was subsequently accepted for publication at BMJ Open.

(This paper received three reviews from its previous journal but only two reviewers agreed to published their review.)

ARTICLE DETAILS

TITLE (PROVISIONAL)	Short email with attachment versus long email without attachment when contacting authors to request unpublished data for a systematic review: a nested randomised trial
AUTHORS	Godolphin, Peter John; Bath, Philip; Montgomery, Alan

VERSION 1 – REVIEW

REVIEWER	Mical Paul Rambam Health Care Campus, Israel
REVIEW RETURNED	20-Jul-2018

GENERAL COMMENTS	This RCT addresses a question of some interest to systematic reviewers, since as stated the quality of the systematic review is sometimes very much dependent on correspondence with primary authors. The main limitation is that the trial is unpowered. I have a few specific comments. • What was the author's hypothesis?• The study sample size was defined in advance by the number of trials in the systematic review. However a power calculation can and should be performed to interpret the lack of difference found between the short and long letter. If I presume that the hypothesis is that a short email will receive more responses and would like to show an increase in response from 30% to 40%, I calculate that the power of the sample is extremely low. Given the lack of power I suggest a short format of a letter or brief report, as a pilot to a larger, powered, trial.• The intervention assessed actually combined two questions: short vs. long email and with/ without attachment. The attachment was part of the short mail only. Many people would be wary of opening an attachment, no matter where it came from, or even to answer a mail with an attachment. To properly assess the question of short vs. long an attachment should have been appended to both emails.• The intervention studied is not the standard request of data from authors. This was a request to actually participate in the systematic review, "to join an international collaboration". Respondents were offered authorship. This should be presented more clearly as the background and in the methods of the study. Short mail with attachment vs. long mail without attachment to receive a response to participate in an international collaboration
--

	performing a systematic review. Authors of systematic review typically send requests for data to authors of the primary studies, without offering them participation in the systematic review or authorship. The scope of this study is completely different.  • Stratification at randomization on the year of the study would have been valuable. The original study years are not well-balanced between the groups. I believe this is an important determinant of response with authors of recent studies more likely to respond than authors of studies published a decade ago. • It would have been difficult to ask for informed consent in this study. However, the email could have included a sentence informing the readers that the correspondence is part of a study evaluating response rates to allow correspondents to say they don't want to participate in a study. I do not perceive this as a problem, since there is no safety issue. There is maybe an autonomy issue, because the number of trials here is small and the authors of the primary studies can be identified. • Although not relevant for this study of emails to authors, the inclusion criteria for the systematic review are not clear. The aim is to investigate "whether central adjudication of the primary outcome in stroke trials has any impact on the main trial primary analysis". But the review includes only "trials of prevention or treatment of stroke that had centrally adjudicated their primary outcome". Why not include trials without central adjudication to be able to compare? • All outcomes can be organized in a single table.
--	---

REVIEWER	Jelte M. Wicherts Tilburg University, the Netherlands
REVIEW RETURNED	23-Jul-2018

GENERAL COMMENTS	In this manuscript, the authors report on a trial to study willingness to share summary results or unpublished data for a systematic review of the potential effect of central adjudication on the primary outcomes in stroke trials. Given empirical results highlighting low rates of sharing such data in the context of systematic reviews and meta-analyses, this work is relevant and timely. The report is quite meticulous and detailed, allowing others to replicate this work in the context of other systematic reviews. However, a concern is that the authors used a relatively small and hence underpowered sample but still appear to accept the null hypothesis in interpreting their results. This is an incorrect use of null hypothesis significance testing. The authors should either rephrase their conclusions ("failed to find a significant difference..." or "found no statistical evidence in favor of a difference...") to reflect this well, or employ appropriate equivalence tests/Bayesian approaches to corroborate the evidence in favor of there being no difference between the trial arms. This issue is important but would not change the gist of the study. It can be dealt with in a revision. Furthermore, I felt that the literature review could be expanded to include some missing references, including the following: Polanin, 2017, J. Clin. Epi, Ohmann et al., 2017, BMJ Open, Hrynaszkiewicz et al., 2016 Res Integr Peer Rev, Teunis et al., 2015 Clin Orthop Relat Res, Fecher et al., 2015, PLOS ONE. It is important that the authors update their literature review and to discuss their work in relation to these earlier works and other similar studies. Besides discussing communities with those earlier
---

	studies, it is also important to acknowledge that in the current study the requesting researchers asked for interest in collaborating instead of asking original researcher to simply share without there being any concrete incentive to do so. This would already heighten sharing rates, and this perhaps warrants more discussion.
--	---

VERSION 1 – AUTHOR RESPONSE

Reviewer: 1

This RCT addresses a question of some interest to systematic reviewers, since as stated the quality of the systematic review is sometimes very much dependent on correspondence with primary authors. The main limitation is that the trial is unpowered. I have a few specific comments.

- What was the author's hypothesis?

Response: The idea for the nested trial arose from discussions about what email format to use when contacting authors of studies eligible for the review. We undertook the nested study from a position of equipoise, with no preconceived idea about whether one email format would be superior to the other. A shorter email may be easier to deal with and quicker to respond to, but it may carry insufficient information (as some people are reluctant about opening attachments) for authors to make a decision. Therefore, authors may prefer an email with full information in the email text and no need to ask for more details. The purpose of our study was to attempt to provide some evidence about which method elicited higher response and a more timely response.

- The study sample size was defined in advance by the number of trials in the systematic review. However a power calculation can and should be performed to interpret the lack of difference found between the short and long letter. If I presume that the hypothesis is that a short email will receive more responses and would like to show an increase in response from 30% to 40%, I calculate that the power of the sample is extremely low. Given the lack of power I suggest a short format of a letter or brief report, as a pilot to a larger, powered, trial.

Response: We have now included text in the statistical analysis section in the Methods. A sample size of 88 trials can detect a difference of ≤ 30 percentage points with 80% power. This assumes that the combined response rate is 50% (which maximises the size of the detectable difference) and therefore, if the true combined response rate differs from 50% then a sample size of 88 would be able to detect a smaller difference with 80% power. For simplicity this ignores any small clustering effect of a few authors with multiple studies.

- The intervention assessed actually combined two questions: short vs. long email and with/ without attachment. The attachment was part of the short mail only. Many people would be wary of opening an attachment, no matter where it came from, or even to answer a mail with an attachment. To properly assess the question of short vs. long an attachment should have been appended to both emails.

Response: Thank you for your comment. The study compared two different methods to elicit response. One was short email with attachment, the other was long email without attachment. We agree that we are unable to draw conclusions solely about email length or attachment due to their combination in the two arms. We have made this clearer in the paper.

- The intervention studied is not the standard request of data from authors. This was a request to actually participate in the systematic review, "to join an international collaboration". Respondents were offered authorship. This should be presented more clearly as the background and in the methods of the study. Short mail with attachment vs. long mail without attachment to receive a response to

participate in an international collaboration performing a systematic review. Authors of systematic review typically send requests for data to authors of the primary studies, without offering them participation in the systematic review or authorship. The scope of this study is completely different.

Response: We have added text in the methods and discussion sections to make it clearer that collaboration (and potential authorship) was offered to participants.

- Stratification at randomization on the year of the study would have been valuable. The original study years are not well-balanced between the groups. I believe this is an important determinant of response with authors of recent studies more likely to respond than authors of studies published a decade ago.

Response: The trial was stratified by year of study (<2011, ≥2011) as detailed in the Randomisation section of the Methods. Furthermore, although stratification did not result in perfect balance between the arms, it was fitted, along with other stratification variables, as a covariate in all multivariable regression models.

- It would have been difficult to ask for informed consent in this study. However, the email could have included a sentence informing the readers that the correspondence is part of a study evaluating response rates to allow correspondents to say they don't want to participate in a study. I do not perceive this as a problem, since there is no safety issue. There is maybe an autonomy issue, because the number of trials here is small and the authors of the primary studies can be identified.

Response: We agree that it is difficult to ask for informed consent in this study and we did consider including a sentence to inform responders that they were in a research study. However, as this intervention is not clinical, and the fact that any mention of this study in correspondence could impact on participant behaviour, we opted not to mention the nested trial to participants. Also, whilst autonomy could be an issue, it would be difficult to identify the participants even if the individual trials are identified (which would be challenging). The primary and secondary contact are never given, only the position on the paper (and IPD for this is never given), therefore, it would be very difficult to identify participants.

- Although not relevant for this study of emails to authors, the inclusion criteria for the systematic review are not clear. The aim is to investigate “whether central adjudication of the primary outcome in stroke trials has any impact on the main trial primary analysis”. But the review includes only “trials of prevention or treatment of stroke that had centrally adjudicated their primary outcome”. Why not include trials without central adjudication to be able to compare?

Response: The inclusion criteria for the systematic review are given in the Prospero review record [13]. Whilst not relevant for this paper, the review includes any stroke trial that had both local on-site assessment and central adjudication of the primary outcome. Therefore, we compare the primary results using unadjudicated data (site-assessed primary outcome) with adjudicated data (centrally adjudicated primary outcome). Including trials that did not adjudicate would not be possible as the comparison is within each trial.

- All outcomes can be organized in a single table.

Response: As tables 2-4 have different column headings, we would prefer to keep these as separate tables for clarity.

Reviewer: 2

In this manuscript, the authors report on a trial to study willingness to share summary results or unpublished data for a systematic review of the potential effect of central adjudication on the primary outcomes in stroke trials. Given empirical results highlighting low rates of sharing such data in the context of systematic reviews and meta-analyses, this work is relevant and timely. The report is quite meticulous and detailed, allowing others to replicate this work in the context of other systematic reviews.

However, a concern is that the authors used a relatively small and hence underpowered sample but still appear to accept the null hypothesis in interpreting their results. This is an incorrect use of null hypothesis significance testing. The authors should either rephrase their conclusions (“failed to find a significant difference...” or “found no statistical evidence in favor of a difference...”) to reflect this well, or employ appropriate equivalence tests/Bayesian approaches to corroborate the evidence in favor of there being no difference between the trial arms. This issue is important but would not change the gist of the study. It can be dealt with in a revision.

Response: Thank you for pointing this out, this was an error on our part. We have edited text in the Discussion section to interpret the data as ‘finding no evidence’ of effects.

Furthermore, I felt that the literature review could be expanded to include some missing references, including the following: Polanin, 2017, J. Clin. Epi, Ohmann et al., 2017, BMJ Open, Hrynaszkiewicz et al., 2016 Res Integr Peer Rev, Teunis et al., 2015 Clin Orthop Relat Res, Fecher et al., 2015, PLOS ONE. It is important that the authors update their literature review and to discuss their work in relation to these earlier works and other similar studies. Besides discussing communities with those earlier studies, it is also important to acknowledge that in the current study the requesting researchers asked for interest in collaborating instead of asking original researcher to simply share without there being any concrete incentive to do so. This would already heighten sharing rates, and this perhaps warrants more discussion.

Response: Thank you for suggestions these useful references. We have expanded our introduction as suggested to include these. We have also included some of these references in the discussion and have lengthened the section in the discussion where we address the fact that we offered collaboration and that this may have increased response rate. Also, we have been clearer in the methods section about offering collaboration (and authorship) as suggested by Reviewer #1.

VERSION 2 – REVIEW

REVIEWER	Jelte Wicherts Tilburg University, The Netherlands
REVIEW RETURNED	04-Oct-2018
GENERAL COMMENTS	In the revision, the authors dealt well with the issues raised earlier by myself and the other reviewer. This work is relevant and well done, and would be an elegant contribution to the literature on data sharing.